# Avian-specific *Salmonella* transition to endemicity is accompanied by localized resistome and mobilome interaction

Chenghao Jia[1], Chenghu Huang[1,2], Haiyang Zhou[1,2], Xiao Zhou[3], Zining Wang[1,2], Abubakar Siddique[1,2], Xiamei Kang[1], Qianzhe Cao[1], Yingying Huang[1,2], Fang He[1,4], Yan Li[1,2], Min Yue[1,2,5,6]*

[1]Department of Veterinary Medicine, Zhejiang University College of Animal Sciences, Hangzhou, China; [2]Hainan Institute of Zhejiang University, Ningbo, China; [3]Ningbo Academy of Agricultural Sciences, Ningbo, China; [4]ZJU-Xinchang Joint Innovation Centre (TianMu Laboratory), Gaochuang Hi-Tech Park, Zhejiang, China; [5]School of Life Science, Hangzhou Institute for Advanced Study, University of Chinese Academy of Sciences, Hangzhou, China; [6]State Key Laboratory for Diagnosis and Treatment of Infectious Diseases, National Clinical Research Center for Infectious Diseases, National Medical Center for Infectious Diseases, The First Affiliated Hospital, College of Medicine, Zhejiang University, Hangzhou, China

*For correspondence:
myue@zju.edu.cn

Competing interest: The authors declare that no competing interests exist.

## eLife Assessment

This **important** study analyzes a large dataset of *Salmonella* gallinarum whole-genome sequences and provides findings regarding the population structure of this avian-specific pathogen. The **convincing** results indicate regional adaptation of the mobilome-driven resistome and a role in the evolutionary trajectory of this pathogen that will interest microbiologists and researchers working on genomics, evolution, and antimicrobial resistance.

**Abstract** Bacterial regional demonstration after global dissemination is an essential pathway for selecting distinct finesses. However, the evolution of the resistome during the transition to endemicity remains unaddressed. Using the most comprehensive whole-genome sequencing dataset of *Salmonella enterica* serovar Gallinarum (*S.* Gallinarum) collected from 15 countries, including 45 newly recovered samples from two related local regions, we established the relationship among avian-specific pathogen genetic profiles and localization patterns. Initially, we revealed the international transmission and evolutionary history of *S.* Gallinarum to recent endemicity through phylogenetic analysis conducted using a spatiotemporal Bayesian framework. Our findings indicate that the independent acquisition of the resistome via the mobilome, primarily through plasmids and transposons, shapes a unique antimicrobial resistance profile among different lineages. Notably, the mobilome-resistome combination among distinct lineages exhibits a geographical-specific manner, further supporting a localized endemic mobilome-driven process. Collectively, this study elucidates resistome adaptation in the endemic transition of an avian-specific pathogen, likely driven by the localized farming style, and provides valuable insights for targeted interventions.

## Introduction

The acquisition of antimicrobial resistance (AMR) by pathogens is well established as one of the most severe threats of the 21st century (*Li et al., 2022a*; *Hou et al., 2023*; *Lee et al., 2023*). It is estimated that over 10 million people could die per year due to AMR by 2050 (*Rappuoli et al., 2017*). Antimicrobial resistance genes (ARGs), collectively known as the resistome, play a pivotal role in AMR development, progression, and amplification (*Larsson and Flach, 2022*). Pathogens typically acquire resistome through mobilomes, like plasmids, via horizontal gene transfer (HGT) (*Tóth et al., 2023*; *Jia et al., 2023*). From an evolutionary perspective, the dynamic change in the resistome is crucial for enhancing pathogen fitness and enabling it to adapt to new ecological niches (*Carr et al., 2020*). Key factors contributing to resistome alterations include disease burden, human activities, climate change, and geographical selection forces (*Wang et al., 2024a*; *Jia et al., 2024a*; *Petersen et al., 2018*; *Wang et al., 2024c*; *Ke et al., 2024*). Understanding the evolutionary trajectory of the AMR and related reservoirs is necessary, as disease management strategies could differ substantially (*Baquero et al., 2021*). However, knowledge regarding the overall profile of pathogen regional adaptation related to the stepwise dynamics of the resistome remains limited.

Before recent advancements in whole-genome sequencing (WGS) technology, little was understood about resistome diversity that affects pathogen endemicity (*Didelot et al., 2012*). Traditional typing methods generally exhibit low resolution, making monitoring and quantitatively comparing horizontal ARG transfer events challenging. The scarcity of appropriate models also poses a limitation. *Salmonella*, a widespread foodborne and zoonotic pathogen with distinct geographical characteristics, has more than 90% of its serovars typically classified as geo-serotypes (*Chen et al., 2024*; *Gossner et al., 2016*). Among the thousands of geo-serotypes, *Salmonella enterica* serovar Gallinarum (*S.* Gallinarum) is an avian-specific pathogen that causes severe mortality, with particularly detrimental effects on the poultry industry in low- and middle-income countries (*Kang et al., 2024b*; *Kang et al., 2022*; *Zhou et al., 2022*). As once a globally prevalent pathogen in the 20th century, *S.* Gallinarum was listed by the World Organization for Animal Health (WOAH) and gradually became an endemic pathogen with sporadic outbreaks following the implementation of eradication programs in most high-income countries, making it a perfect model for study (*Zhou et al., 2023*; *De Carli et al., 2017*).

Nowadays, antimicrobial therapy remains a priority choice against *S.* Gallinarum infections (*Barrow and Freitas Neto, 2011*). The overuse or misuse of antimicrobials has led to increased epidemiological escalation of *S.* Gallinarum, with a regionally higher risk of AMR, especially for sulfonamides, penicillin, and tetracyclines (*Nhung et al., 2017*; *Penha Filho et al., 2016*). To fill the gaps in understanding the evolution of *S.* Gallinarum under regional-associated AMR pressures and its adaptation to endemicity, we collected the most comprehensive set *S.* Gallinarum isolates, consisting of 580 genomes, spanning the period from 1966 to 2023. Using such a unique WGS dataset, we investigated: (1) the population structure and potential geographical transmission history of *S.* Gallinarum at a single base level; (2) the dynamic resistome and mobilome changes that are associated with *S.* Gallinarum transition to an endemic variant; and (3) horizontal resistome transfer frequency and pattern between distinct regions.

## Results

### Global distribution of *S.* Gallinarum links with distinct sublineages

To understand the global geographical distribution and genetic relationships of *S.* Gallinarum, we assembled the most comprehensive *S.* Gallinarum WGS dataset (n=580, *Supplementary file 1*, *Supplementary file 2*), comprising 535 publicly available genomes and 45 newly sequenced genomes (*Jia et al., 2024b*). The core-genome single nucleotide polymorphism sites (cgSNPs) were obtained by using the fully sequenced genome of *S.* Gallinarum R51 as a reference. Through hierarchical Bayesian analysis of cgSNPs, it was confirmed that *S.* Gallinarum divides into three biovars: *S.* Gallinarum biovar Pullorum (bvSP) (n=528/580, 91.03%), *S.* Gallinarum biovar Gallinarum (bvSG) (n=50/580, 8.6%), and *S.* Gallinarum biovar Duisburg (bvSD) (n=2/580, 0.34%). An association was identified between the biovar type of *S.* Gallinarum and its global geographical distribution. Most bvSP isolates were from Asia (436/528), with fewer occurrences in Europe (54/528), South America (12/528), and North America (5/528). In bvSG, South America (18/50, 36%) is the primary source of isolates, followed by

North America (9/50, 18%), Europe (9/50, 18%), Africa (7/50, 14%), and Asia (3/50, 6%) (*Figure 1— figure supplement 1*).

From a lineage perspective, bvSP can be further classified into five lineages: L1 (10/528), L2a (45/528), L2b (163/528), L3b (169/528), and L3c (141/528). Importantly, L3a, previously considered distinct, has been revealed as a sublineage of L3c. The predominant lineage types vary across different continents. Regarding predominant bvSP sources, Asia stands out, with L3c, L3b, and L2b being the main lineage types. On the other hand, L3b and L2a are more widespread in Europe and America, respectively (*Figure 1a*). For the bvSP strains from Asia included in our dataset, we found that all originated from China. To further investigate the distribution of bvSP across different regions in China, we categorized them into three distinct regions: southern, eastern, and northern (*Supplementary file 3*). Our findings indicate significant variations in bvSP across various regions from geo-temporal aspects. Prevalence was highest in the eastern region (233/436), with lower incidences in the northern (137/436) and southern regions (65/436). Interestingly, the predominant bvSP lineage type also varied. In eastern and southern China, L2b and L3c exhibited as the predominant lineage types, while in northern China, L3b and L3c held nearly equivalent positions (*Figure 1b*).

An analysis of the temporal prevalence of bvSP in China revealed a gradual replacement pattern of the lineages. Before the 2000s, the predominant lineage type was L3b. Then, from the 2000s to the 2010s, L3b and L1 were continually replaced by L3c, with L2b becoming the predominant lineage after the 2010s (*Figure 1c*). However, the replacement pattern varied among different regions. Specifically, the same pattern was observed in eastern China, but for northern and southern China, only replacements from L3b to L2b and L1 to L3c were observed (*Figure 1—figure supplement 2*). These findings might indicate accelerated localized adaptation of bvSP.

## Genomics portrait bvSP historic transmission

Considering the historical pandemic status of bvSP, we then investigated its global geographical transmission routes to understand its evolution into an endemic pathogen. L2b and L3b were identified as the dominant global lineages due to their AMR risks and potential intercontinental transmission events. Other lineages were also analyzed using a Bayesian model with a relaxed molecular clock to infer their historical evolution (*Figure 2—figure supplements 1–3*). The temporal structure was verified (*Figure 2—figure supplement 4*). Our findings show that the origin of L3b in China can be traced back to as early as 1683 (95% CI: 1608–1839). In contrast, the earliest possible origin of L2b in China dates back to 1880 (95% CI: 1838–1902) (*Figure 2*). Then, we specifically estimated the time points for the first intercontinental transmission events for the two major lineages. Our results indicate that L2b likely underwent two major intercontinental transmission events. The first occurred around 1893 (95% CI: 1870–1918), with transmission from China to South America. The second major transmission event occurred in 1923 (95% CI: 1907–1940), involving the spread from South America to Europe. In contrast, the transmission pattern of L3b, an *S*. Gallinarum lineage originating in China, appears relatively more straightforward. It underwent only one intercontinental transmission event, from China to Europe, likely around 1790 (95% CI: 1661–1890) (*Figure 2—figure supplement 5*).

## Endemic isolates further support localized dissemination by bvSP

To investigate the dissemination pattern of bvSP, we obtained 45 newly isolated bvSP from 734 samples (6.1% overall isolation rate) collected from diseased chickens at two farms in Yueqing and Taishun, Zhejiang Province. Those bvSP can be classified into two dominant sequence types (STs): ST3717 and ST92. Isolates from Yueqing were all identified as ST3717, while most Taishun isolates were characterized as ST92. Only the E404 isolate was identified as ST2151 (*Figure 3a*). Of note, a lineage-preferential association was also observed, with bvSP isolated from Taishun belonging to L3b and those from Yueqing belonging to L2b. To further understand the genetic relationship among isolates in the two regions, we calculated cgSNP distances between isolates. Isolates with cgSNP distances less than two were determined as the threshold to be involved in the same transmission event. Interestingly, genetic clustering revealed that isolates from the same farm exhibited a significantly close genetic correlation (*Figure 3b*). No historical transmission events of bvSP were found between Taishun and Yueqing.

Other genetic characteristics also exhibited a similar pattern. The ARG and plasmid profile disclosed that *IncFII(S)* and *ColpVC* plasmids exhibited the highest carriage percentages (100%) among the 45

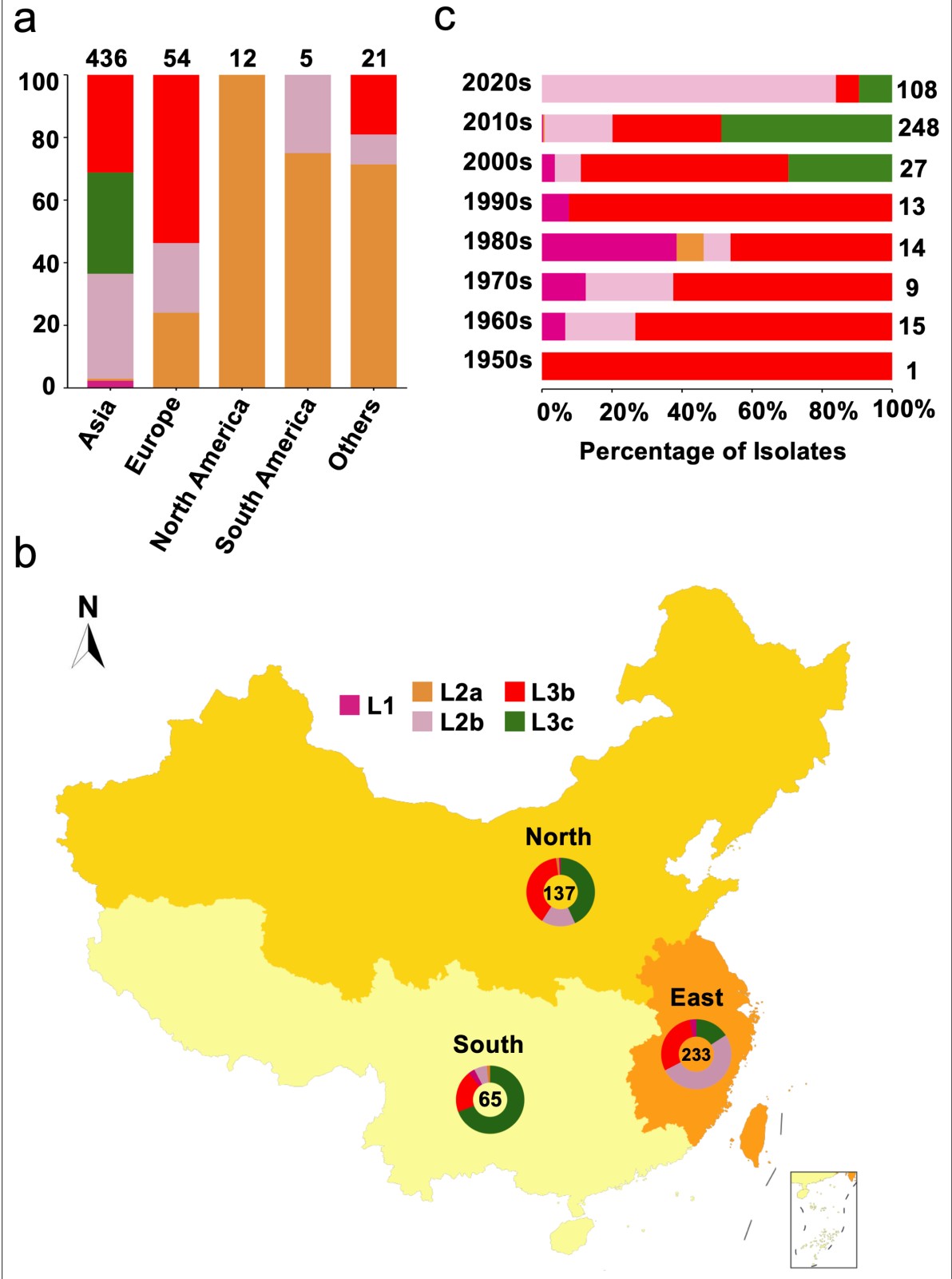

**Figure 1.** Genetic diversity of *S.* Gallinarum biovar Pullorum (bvSP) by geography and time. Different colors were used to represent the various lineages of bvSP: fuchsia for L1, orange for L2a, pink for L2b, red for L3b, and green for L3c. (**a**) The bvSP in the dataset is classified into five continents based on their isolation regions. The bar graph illustrates the distribution of lineage-specific bvSP across continents, depicted as percentages. The total sample size for each bar is indicated at the top. (**b**) Geographical distribution of bvSP isolated from China. Doughnut charts in the map show the proportion of

*Figure 1 continued on next page*

Figure 1 continued

lineage types of bvSP collected in the corresponding region, with the total number of isolates in the center. (**c**) The bar graph shows the distribution of bvSP isolated in China by lineage per decade. The total sample size for each bar is also indicated on the right side.

The online version of this article includes the following figure supplement(s) for figure 1:

**Figure supplement 1.** The evolutionary structure of global *S*. Gallinarum.

**Figure supplement 2.** The primary prevalence of *S*. Gallinarum biovar Pullorum (bvSP) lineages varies across different regions of China over time, with consideration given to the eastern, northern, and southern regions.

bvSP isolates. The *IncX1* plasmid was exclusively identified in the E404 isolate. Furthermore, E404 harbored two distinctive resistance genes, $bla_{TEM-1B}$ and *sul2*, in contrast to the absence of ARGs in the remaining bvSP isolates. We also estimated the invasiveness index to assess the invasiveness of isolated bvSP and showed that bvSP from the same farm had comparable invasive abilities. However, isolates from Taishun demonstrated a higher invasive ability than those from Yueqing (*Figure 3c*, *Supplementary file 4*).

Furthermore, we simulated potential transmission events between the bvSP strains isolated from Zhejiang Province (n=95) and those from China with available provincial information (n=435) using two SNPs as the threshold. As a result, the transmission events showed a strong geographic-preferential distribution. We identified a total of 53 potential transmission events, all of which occurred exclusively within Zhejiang Province. No inter-provincial transmission events were detected, supporting the statement that bvSP in China is a highly localized pathogen (*Figure 3—figure supplement 1*, *Supplementary file 5*).

## Mobilome and resistome distinct lineage-preferential association in a regional manner

The mobilome drives partitions of the resistome and plays a pivotal role in shaping the ecological niches and local adaptation of bacteria. Therefore, a quantitative evaluation of the resistome and mobilome is necessary to enhance the understanding of the localized distribution of bvSP. For *S*. Gallinarum, a total of 13 classes of ARGs, which can be further classified into six categories, were identified. The results revealed that bvSP exhibited a significantly greater resistome than bvSG and bvSD (*Figure 4a*). Among them, *sul2* (196/528, 37.1%) has the highest prevalence, followed by $bla_{TEM-1B}$ (183/528, 34.7%) and *tet(A)* (104/528, 19.7%). Moreover, the resistome also demonstrated a lineage-preferential association. We observed that L3b is more inclined to carry $bla_{TEM-1B}$ and *sul2* than other lineages, whereas *tet*(A) exclusively exists within L3 (*Supplementary file 6*). The diversity of resistome carried by L3 may elevate the risk of multidrug resistance (MDR). Notably, AMR risks vary among different regions of China, with the highest risk observed in southern China, followed by the northern and eastern regions. Interestingly, *tet*(A), $bla_{TEM-1B}$, and *sul2* were predominant resistome types prevalent across all regions of China. Meanwhile, *aadA5* was typically more commonly found in southern China (*Figure 4b and c*).

Four categories of the mobilome—prophage, plasmid, transposon, and integron—were characterized, revealing strong lineage-specific patterns (*Figure 4—figure supplement 1a*, *Supplementary file 7*). In silico analyses revealed prophages emerge as the most diverse mobilome type, with *Entero_mEp237*, *Escher_500465_1*, and *Klebsi_4LV2017* being more likely associated with L1, L2a, and L2b, respectively, while the carrier percentage of *Escher_pro483* is significantly higher among L3c. Interestingly, *Salmon_SJ46*, *Gifsy_2*, *Escher_500465_2*, and *Shigel_SfIV* were consistently observed across all bvSP lineages and were found to be duplicated on the chromosome. For plasmids, *IncFII(S)* was predominantly carried across all bvSP lineages (520/528, 98.5%) (*Figure 4—figure supplement 1b*). Notably, *ColpVC* had the lowest carriage percentage among L1 (2/10, 20%), whereas *IncX1* was explicitly associated with L3c (140/141, 99.3%).

The diversity of transposons and integrons is comparatively lower in bvSP. Specifically, we identified seven types of transposons and four types of integrons. Transposons were found to be more abundant, comprising a total of 255 in 528 bvSP. Among them, *Tn801* (183/528, 35%) had the highest carriage rate, followed by *Tn1721* (48/528, 9.1%). Interestingly, L3c is more likely to carry *Tn801* and *Tn1721* transposons than other lineages. In contrast, a total of 41 integrons were identified among the 528 bvSP, all of which belong to class I: *In498* (n=21), *In1440* (n=14), *In473* (n=4), and *In282* (n=2).

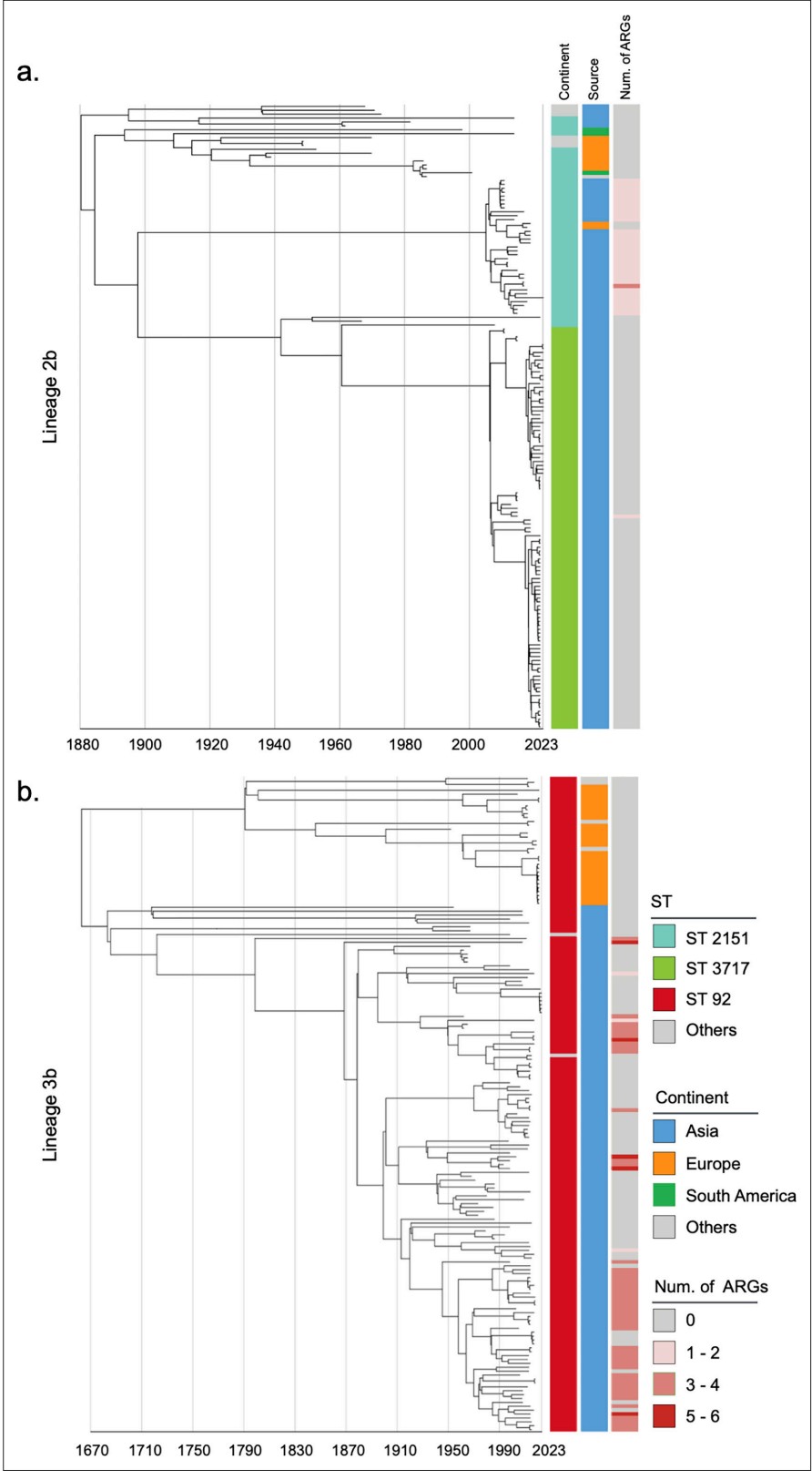

**Figure 2.** Phylogenetic tree of *S.* Gallinarum L2b and L3b based on a spatiotemporal Bayesian framework. The phylogenetic tree on the left was constructed using a reference-mapped multiple core-genome SNPs sequence alignment, with recombination regions detected and removed by Gubbins. The spatiotemporal Bayesian framework was configured with the 'GTR' substitution model, 4 Gamma Category Count, 'Relaxed Clock Log

*Figure 2 continued on next page*

*Figure 2 continued*

Normal' model, 'Coalescent Bayesian Skyline' tree prior model, and a Markov Chain Monte Carlo (MCMC) chain length of 100,000,000, with sampling every 10,000 iterations. Convergence was assessed using Tracer, ensuring all parameter effective sampling sizes (ESS) exceeded 200. Evolutionary time is represented by the length of the branches. The heatmap on the right displays, respectively, the sequence type (ST), region of isolation, and the number (Num.) of antimicrobial resistance genes (ARGs) carried by the corresponding *Salmonella* Gallinarum. (**a**) indicates the phylogenetic tree for Lineage 2b, (**b**) indicates the phylogenetic tree for Lineage 3b.

The online version of this article includes the following figure supplement(s) for figure 2:

**Figure supplement 1.** Phylogenetic tree of *Salmonella* Gallinarum L1 based on a spatiotemporal Bayesian framework.

**Figure supplement 2.** Phylogenetic tree of *Salmonella* Gallinarum L2a based on a spatiotemporal Bayesian framework.

**Figure supplement 3.** Phylogenetic tree of *Salmonella* Gallinarum L3c based on a spatiotemporal Bayesian framework.

**Figure supplement 4.** Assessment of the temporal structure (L1-L3c).

**Figure supplement 5.** Historical international transmissions of *S.* Gallinarum biovar Pullorum (bvSP) lineages L2b and L3b are depicted with arrows representing the transmission paths. The pink and red lines represent L2b and L3b, respectively.

**Figure supplement 6.** Recombination removal using Gubbins.

The integrons also exhibited a lineage-preferential distribution, predominantly carried by L3b. From a geographical distribution perspective, bvSP from northern and southern China carried the most extensive mobilome, followed by those from eastern China. Interestingly, bvSP isolated from northern China typically have more diverse mobilome types (*Figure 4—figure supplement 2*).

## Plasmid and transposon guide resistome geo-temporal dissemination

Variations in regional antimicrobial use may result in uneven pressure for selecting AMR. The mobilome is considered the primary reservoir for spreading resistome, and a consistent trend between the resistome and the mobilome has been observed across different lineages, from L1 to L3c. We observed an overall gradual rise in the resistome quantity carried by bvSP across various lineages, correlating with the total mobilome content (*Figure 5—figure supplement 1*). Furthermore, we investigated the interplay between particular mobile elements and resistome types in bvSP.

In bvSP, the predominant resistome is mainly associated with certain types of mobilome (*Figure 5a*). Specifically, regarding $bla_{TEM\_1B}$, $tet(A)$, and $sul1$, which are highly prevalent among bvSP, we found most $bla_{TEM\_1B}$ were carried by the transposon *Tn801* and *Tn1721*, the plasmid *IncX1*, and the prophage SJ46. Additionally, $tet(A)$ was primarily associated with the transposon *Tn1721* and the plasmid IncX1. Integrons also facilitated the dissemination of the resistome. For instance, *In498* and *In1440* facilitate the dissemination of $sul1$, leading to quinolone resistance. Our results indicate that plasmids and transposons are the predominant reservoirs for the resistome in bvSP, with prophages and integrons following closely behind (*Figure 5b*). Interestingly, we observed that the primary reservoir for the resistome shows regional variations. In the southern and northern regions of China, the primary reservoirs for $bla_{TEM\_1B}$ are *Tn801*, *Tn1721*, *IncX1*, and prophage SJ46. However, *Tn1721* was less frequently found in bvSP, which carries $bla_{TEM-1B}$ from the eastern region. Similarly, the reservoir for $tet(A)$ exhibits greater diversity in the east region but less diversity in the southern region (*Figure 5c–e*).

## Horizontal transfer of resistome occurs widely in localized bvSP

Horizontal transfer of the resistome facilitates the acquisition of AMR among bacteria, which may record the distinct acquisition event in the bacterial genome. To compare these events in a geographical manner, we further investigated the HGT frequency of each ARG carried by bvSP isolated from China and explored the HGT frequency of resistome between three defined regions. Potentially horizontally transferred ARGs were defined as those with perfect identity (100% identity and 100% coverage) and were located on mobilome genetic elements (MGEs) across different strains (*Figure 6a*). We first categorized a total of 621 ARGs carried by 436 bvSP isolates collected in China and found that 415 ARGs were located on MGEs (*Supplementary file 8*). After excluding the ARGs not associated with

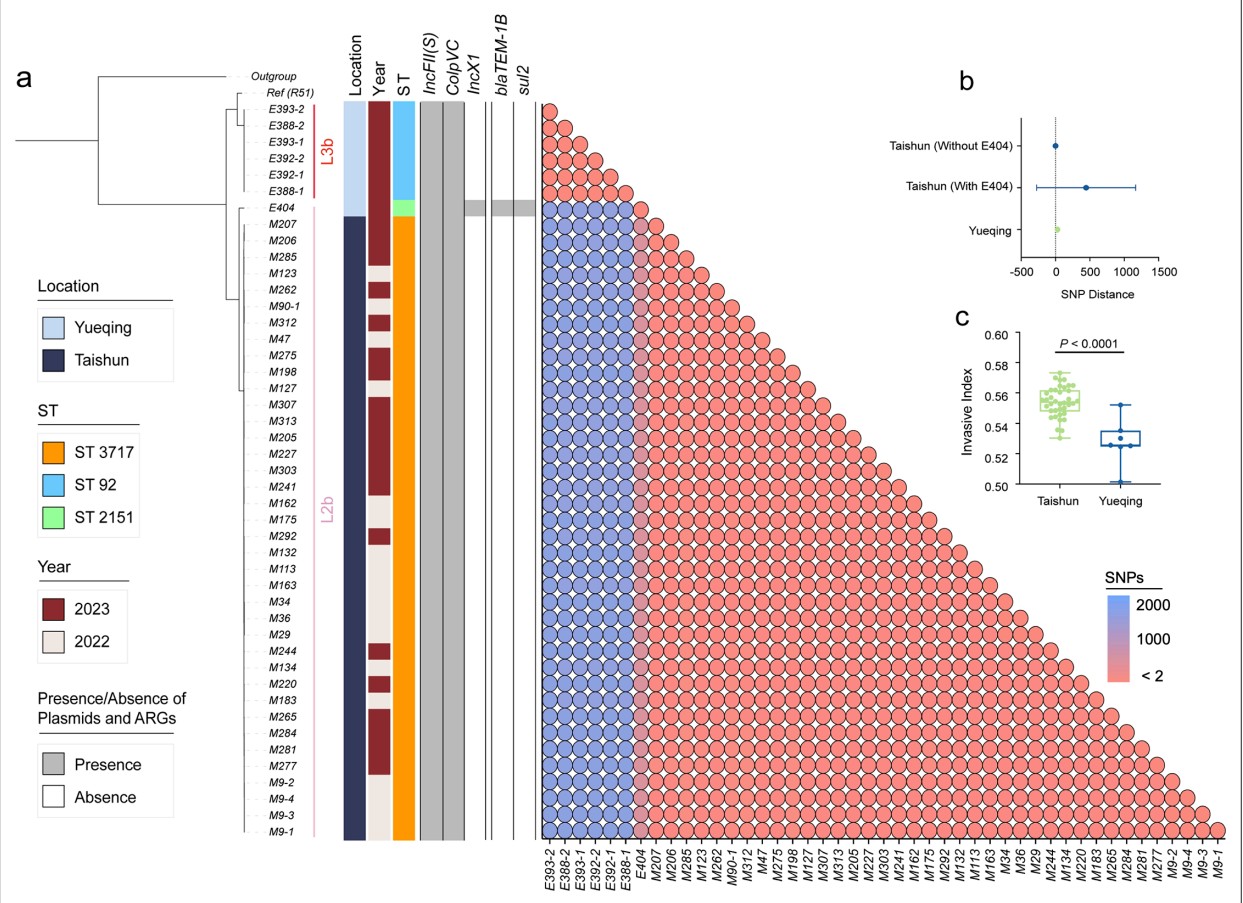

**Figure 3.** Genomic characteristics of 45 newly isolated *S.* Gallinarum biovar Pullorum (bvSP) strains from Taishun and Yueqing, Zhejiang Province, China. (**a**) The phylogenetic tree, constructed using core-genome single nucleotide polymorphism sites (cgSNPs), categorizes the 45 bvSP strains into two distinct lineages (L3b and L2b). On the left side of the heatmap, information on isolation regions, sequence types (STs), and sampling times are displayed, with various colors indicating different categories as specified on the left. The right section of the heatmap presents a detailed matrix showing plasmids, antimicrobial resistance genes (ARGs), and cgSNP distances. The presence of a plasmid or ARG in an isolate is denoted by gray shading, while absence is indicated by white. cgSNPs below two were used as the threshold, with red circles signifying a higher probability of transmission between isolate pairs. (**b**) The average cgSNP distance between isolates from Taishun and Yueqing. E404 led to an increase in the mean cgSNP distance of bvSP from Taishun. (**c**) Invasiveness index of bvSP in Taishun and Yueqing. The results show a higher invasiveness index for bvSP isolated from Taishun, indicating that bvSP isolated from Taishun might have greater invasive capabilities among vulnerable hens.

The online version of this article includes the following figure supplement(s) for figure 3:

**Figure supplement 1.** Potential transmission events (n=53) of *S.* Gallinarum biovar Pullorum (bvSP).

MGEs, our findings reveal that HGT of ARGs is widespread among Chinese bvSP isolates, with an overall transfer rate of 92%. Specifically, 50% of the ARGs exhibited an HGT frequency of 100%. Using the HGTphyloDetect pipeline, we verified the HGT transmission potential for each ARG sequence. The results demonstrated that ARGs with an HGT frequency greater than zero (*bla*TEM-1B, *sul1*, *dfrA17*, *aadA5*, *sul2*, *aph(3'')-Ib*, *tet(A)*, *aph(6)-Id*) all had Alien Index (AI) scores exceeding 45 and out_perc values greater than 90, indicating that these ARGs may have undergone extensive horizontal transfer events (*Figure 6b*, *Supplementary file 9*). It is noteworthy that certain resistance genes, such as *tet(A)*, *aph(3'')-Ib*, and *aph(6)-Id*, appear to be less susceptible to horizontal transfer.

However, different regions generally exhibited a considerable difference in resistome HGT frequency. For specific ARG type, we found *tet(A)* is more prone to horizontal transfer in the southern region, and this proportion was considerably lower in the eastern region. Interestingly, certain ARGs, such as *aph(6)-Id*, undergo horizontal transfer only within the eastern and northern regions of China (*Figure 6c*). Notably, as a localized transmission pathogen, resistome carried by bvSP exhibited a

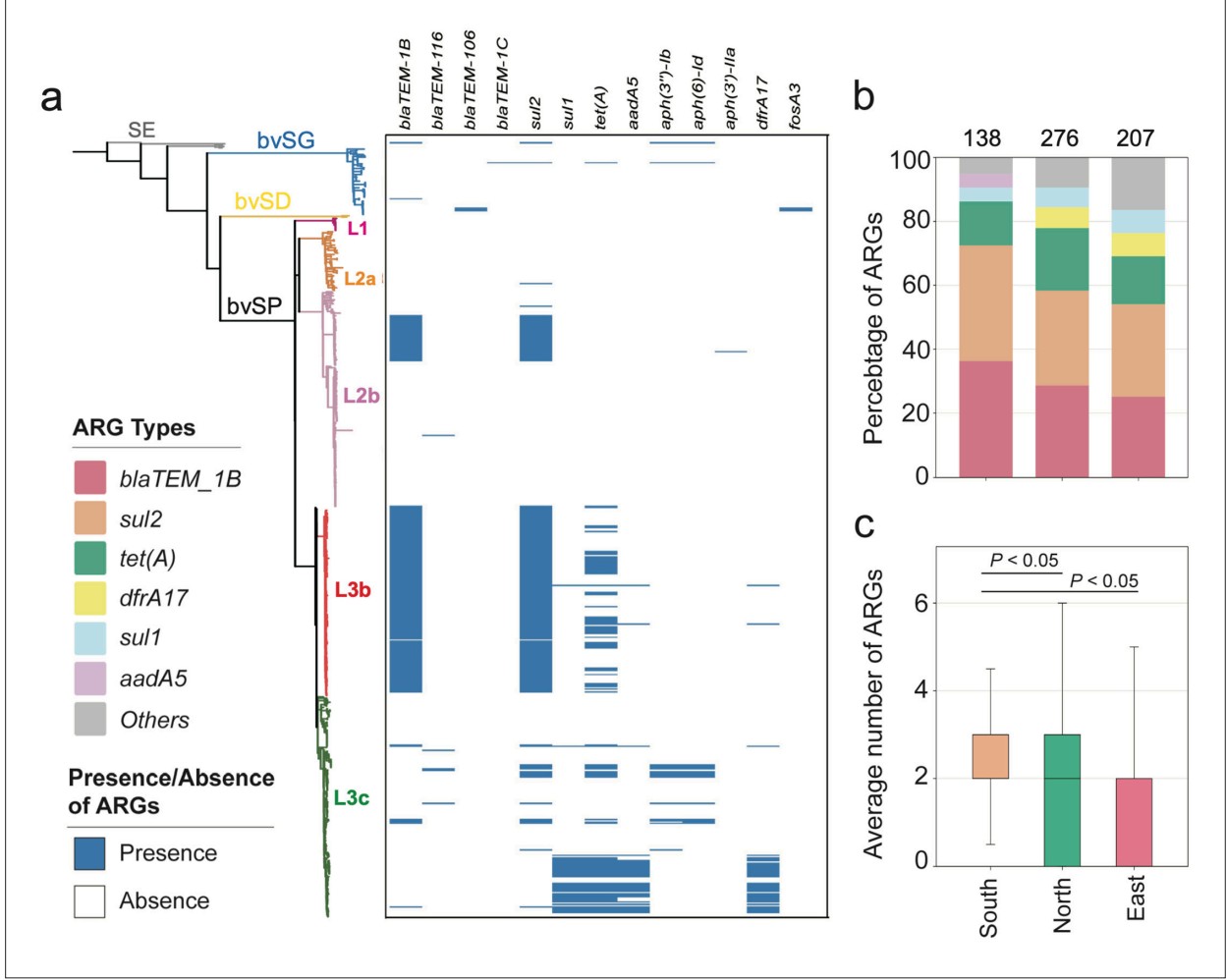

**Figure 4.** The antimicrobial resistance genes (ARGs) carried by *S*. Gallinarum. (**a**) The phylogenetic tree was constructed using core-genome single nucleotide polymorphism sites (cgSNPs), revealing three *S*. Gallinarum biovars: *S*. Gallinarum biovar Pullorum (bvSP), *S*. Gallinarum biovar Duisburg (bvSD), and *S*. Gallinarum biovar Gallinarum (bvSG). Additionally, *Salmonella* serovar Enteritidis (SE) is represented by a gray line. Further, bvSP can be subdivided into five lineages: fuchsia (L1), orange (L2a), pink (L2b), red (L3b), and green (L3c). The heatmap on the right indicates the resistome carried by the corresponding *Salmonella*. (**b**) The dominant resistome types in different regions of China. The y-axis represents the percentage of each dominant resistome. The total sample size for each bar is indicated at the top. (**c**) The average number of resistome carried by bvSP is from different regions of China.

The online version of this article includes the following figure supplement(s) for figure 4:

**Figure supplement 1.** The carriage of four predominant mobilome.

**Figure supplement 2.** Types of predominant mobile genetic elements carried by various regions.

dynamic potential among interregional and local demographic transmission, especially from northern region to southern region (HGT frequency = 93%) (*Figure 6d*, *Supplementary file 10*).

## Discussion

Over the past decades, the emergence of AMR has garnered significant attention from both public health agencies and the poultry industry (*Huang et al., 2024*). Region-associated pressures shaped the dissemination patterns of bacterial pathogens, transitioning from pandemic to endemic, and led to dynamic changes in the resistome. Here, utilizing *S*. Gallinarum as a prototype for an endemic pathogen that once spread globally and assembling the most comprehensive *S*. Gallinarum WGS database, we have discovered that region-associated selection pressures could influence the AMR

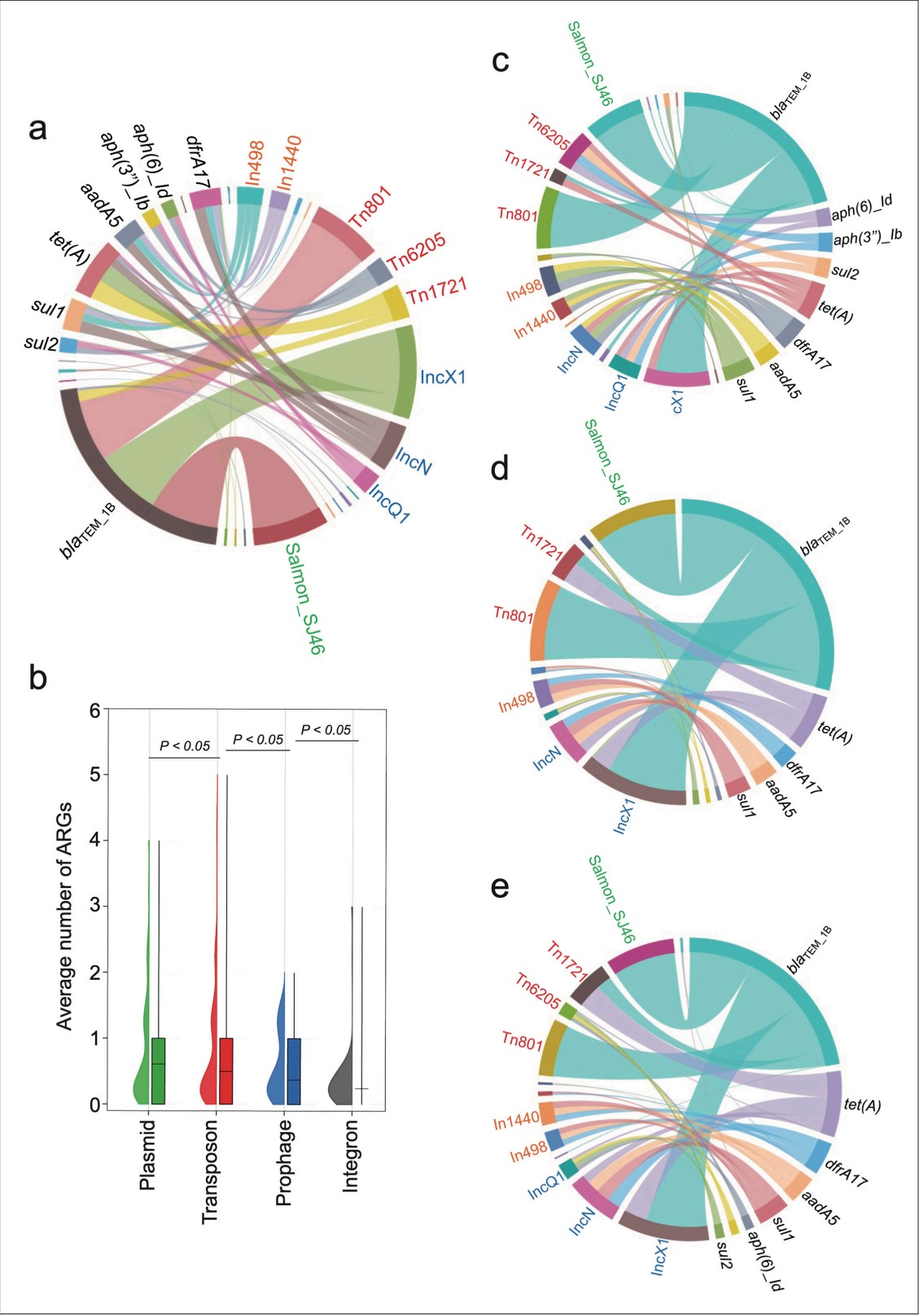

**Figure 5.** The primary source of resistome is carried by distinct mobilome. Different font colors denote various mobilome types. Specifically, orange signifies integrons, red denotes transposons, blue represents plasmids, and green indicates prophages. A black font is utilized to distinguish the categories of resistome. The connecting line between the resistome and the mobilome represents the potential carrying relationship. (a) The mobilome-carried antimicrobial resistance genes (ARGs) among *S.* Gallinarum biovar Pullorum (bvSP). (b) The average number of ARGs carried by the

*Figure 5 continued on next page*

*Figure 5 continued*

four mobilome genetic elements (MGEs) in bvSP. The unpaired *t*-test was used to compare the differences between isolates, with p<0.05 considered statistically significant. (**c–e**) The mobilome-carried ARGs among bvSP isolated from the eastern, southern, and northern regions of China.

The online version of this article includes the following figure supplement(s) for figure 5:

**Figure supplement 1.** Trends in both resistome and mobilome quantities over time and across lineages.

risks of *S.* Gallinarum within lineage- and region-specific patterns. These pressures might also influence lineage-distinct evolutionary structures and transmission histories of *S.* Gallinarum.

The biovar types of *S.* Gallinarum have been well defined as bvSP, bvSG, and bvSD historically (***Kisiela et al., 2005***). Among these, bvSP can be further subdivided into five lineages (L1, L2a, L2b, L3b, and L3c) using hierarchical Bayesian analysis. Different sublineages exhibited preferential geographical distribution, with L2b and L3b of bvSP being predominant global lineage types with a high risk of AMR. The historical geographical transmission was verified using a spatiotemporal Bayesian framework. The result shows that L3b was initially spread from China to Europe in the 18th to 19th century, which may be associated with the European hen fever event in the mid-19th century (***Burnham, 1855***). L2b, on the other hand, appears to have spread to Europe via South America, potentially contributing to the prevalence of bvSP in the United States. Considering the economic losses caused by bvSP, the United States, Europe, and other industrialized countries implemented eradication programs in the mid-19th century, thus, eliminating the risk of Pullorum disease. Implementing similar measures is challenging due to China's vast geographical area and various economic factors, resulting in bvSP becoming an endemic pathogen, mainly in China.

Additionally, 45 new bvSP isolates were obtained from two different locations (Taishun and Yueqing, Zhejiang Province). The cgSNP distances (<2 SNPs) revealed potential transmission events occurring exclusively among isolates from the same location. Traceback analysis using isolates from Zhejiang Province and other Chinese isolates with available provincial information further supports that bvSP in China is a highly localized pathogen. Moreover, the invasiveness index of bvSP from Taishun and Yueqing suggests that different lineages of *S.* Gallinarum recovered from distinct regions may exhibit biological differences. Previous studies have shown that strains with higher invasiveness indexes tend to be more virulent in hosts (***Cuypers et al., 2023***; ***Pulford et al., 2021***), potentially causing neurological or arthritic symptoms in *S.* Gallinarum infections. Furthermore, strains with varying invasiveness indexes have been confirmed to differ in their biofilm formation abilities and metabolic capacities for carbon compounds (***Van Puyvelde et al., 2019***).

Humanity appeared powerless against epidemics until the advent of antimicrobials, which are still the primary agents used to combat *S.* Gallinarum infection (***Worboys, 2000***; ***Porter, 1997***). As a manifestation of localization-related forces, the emergence of AMR allows pathogens to acquire additional fitness benefits (***Helekal et al., 2023***). Our findings further indicate that the mobilome facilitates the acquisition of AMR among bvSP, mainly through plasmids and transposons, such as *IncX1*, *IncQ1*, and *IncN* type plasmids, as well as *Tn801*, *Tn6205*, and *Tn721* type transposons. A survey conducted in China between 1962 and 2010 documented the association of class 1 integrons with AMR among bvSP (***Gong et al., 2013***). Our study further demonstrates that *In498*, *In1440*, *In473*, and *In282*, all belonging to class 1 integrons, play a significant role in carrying ARGs associated with sulfonamides and trimethoprim. However, compared to other *Salmonella* serovars (***Li et al., 2022c***; ***Teng et al., 2022***; ***Pan et al., 2022***), the level of resistome in bvSP remains lower.

Considering lineage prospects, recently emerged lineages present a higher risk of AMR, with a lineage-specific distribution of resistome. In L2b and L3c, the most prevalent groups of ARGs include $bla_{TEM-1b}$ and *sul2*. L3c, which exhibits the highest resistome richness with *sul1*, *tet(A)*, *aadA5*, and *dfrA17*, has a higher likelihood of MDR. However, the transmission routes of resistome showed a high degree of consistency across continents. Globally, a small proportion of resistome carriers belonging to L2b and L3b were observed in the Americas and Europe, while isolates from Asia exhibited a notable rise in resistome carriage. Therefore, we speculate that the mobilome in bvSP may have been acquired locally in Asia, possibly due to increased antimicrobial usage and an intensive industrialized poultry farming pattern within the regional poultry industry (***Kumar et al., 2019***). The L3c, being a local Asian lineage and more prone to acquiring additional resistome locally, provided further support.

In China, the endemic region of Asia, an unexpected rising trend has been observed from the 1970s to the 2020s. However, the risk of AMR varies among distinct regions (***Jin et al., 2024a***;

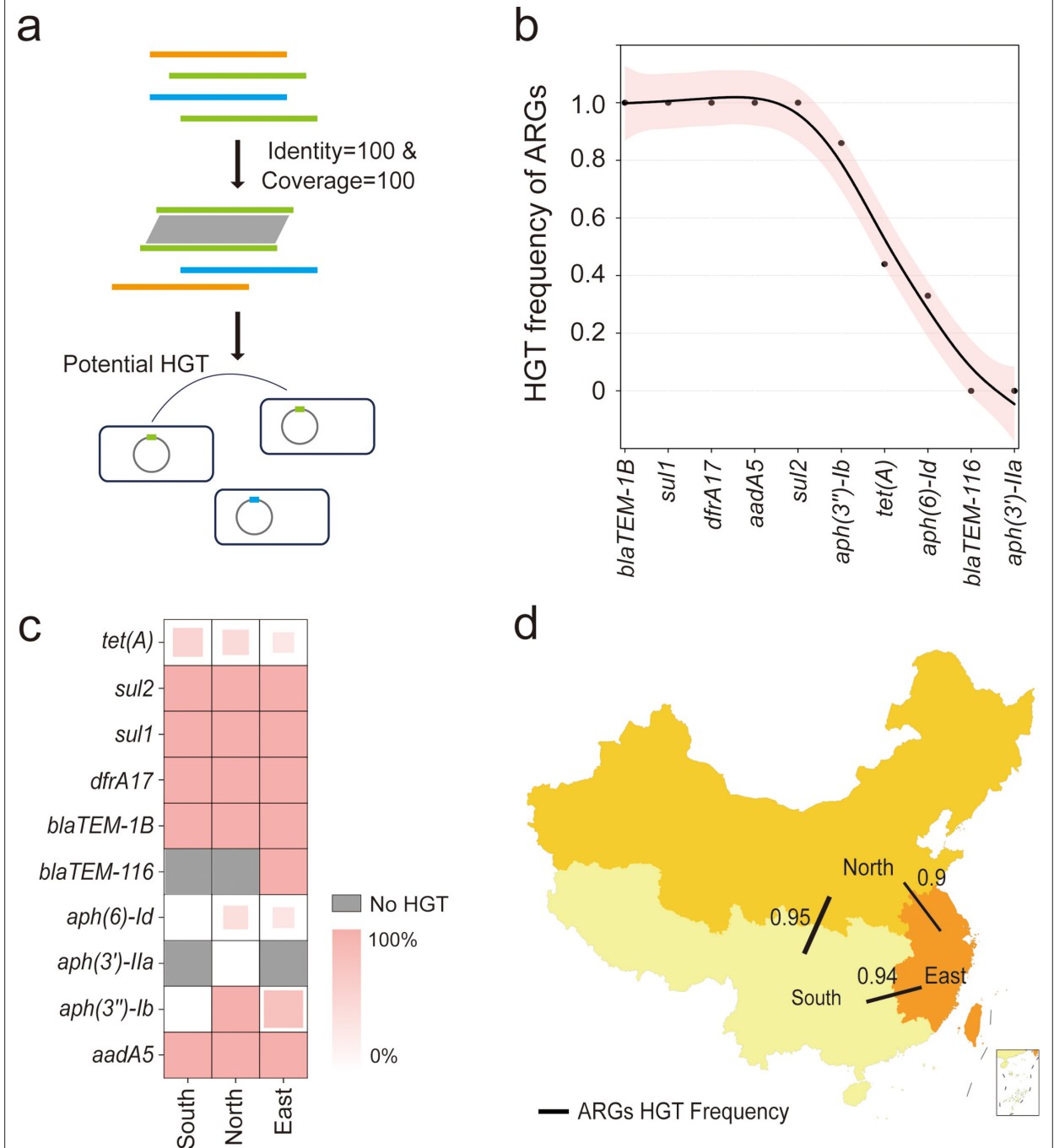

**Figure 6.** The horizontal gene transfer (HGT) frequency of the resistome among *S*. Gallinarum biovar Pullorum (bvSP) isolated from China. (**a**) Workflow for identification of horizontally transferred antimicrobial resistance genes (ARGs) in *Salmonella*. (**b**) The x-axis represents the resistome of bvSP, while the y-axis represents the corresponding levels of HGT frequency. (**c**) The HGT frequency level of specific ARGs carried by bvSP isolated from various regions of China. Deeper colors mean higher HGT frequency. (**d**) The frequency of horizontal retransmission of the resistome between different regions of China. A higher value indicates more frequent transfer events of resistome between two regions. The colors on the map represent the number of bvSP isolates in each region. Darker colors indicate a higher number of bvSP isolates in that area.

*Elbediwi et al., 2021*; *Li et al., 2022b*; *Xu et al., 2020*). Overall, the average number of ARGs carried by bvSP isolated from eastern China was the lowest, while it was higher among bvSP isolated from northern and southern China. This variation may be influenced by differences in the economy, climate, or farming practices among different regions. Interestingly, the widespread horizontal transfer of the

resistome across regions indicates that the resistome has a different mode of transmission compared to its pathogenic bacteria. Specifically, as a localized pathogen, our results show a low frequency of cross-regional transmission of bvSP, but up to 90% of ARGs are transmitted cross-regionally. This risk has not been previously observed in *S*. Gallinarum, suggesting that potential intermediate hosts may play a key role.

In summary, the findings of this study highlight that *S*. Gallinarum remains a significant concern in developing countries, particularly in China. Compared to other regions, *S*. Gallinarum in China poses a notably higher risk of AMR, necessitating the development of additional therapies, i.e., vaccine, probiotics, bacteriophage therapy in response to the government's policy aimed at reducing antimicrobial use (*Jin et al., 2024b*; *Golkar et al., 2014*). Furthermore, given the dynamic nature of *S*. Gallinarum risks across different regions, it is crucial to prioritize continuous monitoring in key areas, particularly in China's southern regions where extensive poultry farming is located. Lastly, from a One-Health perspective, controlling AMR in *S*. Gallinarum should not solely focus on local farming environments, with improved overall welfare on poultry and farming style. The breeding pyramid of industrialized poultry production should be targeted at the top, with enhanced and accurate detection techniques (*Kang et al., 2024a*). More importantly, comprehensive efforts should be made to reduce antimicrobial usage overall and mitigate potential AMR transmission from environmental sources or other hosts (*Peng et al., 2022*; *Siddique et al., 2024*; *Jiang et al., 2023*; *Jiang et al., 2025*; *Wang et al., 2025*).

However, the current study has some limitations. First, despite assembling the most comprehensive WGS database for *S*. Gallinarum from public and laboratory sources, there are still biases in the examined collection. The majority (438/580) of *S*. Gallinarum samples were collected from China, possibly since the WGS is a technology that only became widely available in the 21st century. This makes it impractical to sequence it on a large scale in the 20th century, when *S*. Gallinarum caused a global pandemic. So, we suspect that human intervention in the development of this epidemic is the main driving force behind the fact that most of the strains in the dataset originated in China. In our future work, we aim to actively gather more data to minimize potential biases within our dataset, thereby improving the robustness and generalizability of our findings. Second, in silico analysis relies heavily on sophisticated, continuously updated databases. Although we utilized the most up-to-date databases, the exact prevalence of MGEs and ARGs may be underestimated. Nevertheless, by using the most extensive global WGS data from *S*. Gallinarum, we elucidate a dynamic resistome evolution framework in a single bacterial pathogen from pandemic to endemic, which is hallmarked by a gradual increase in AMR risks and a stepwise resistome adaptation. The insights gained in this study will be helpful in further prevention and surveillance of the AMR risks of localized evolving pathogens.

## Materials and methods
### Bacterial isolates

All 734 samples of dead chicken embryos aged 19–20 days were collected from Taishun and Yueqing in Zhejiang Province, China. After a thorough autopsy, the liver, intestines, and spleen were extracted and added separately into 2 mL centrifuge tubes containing 1 mL PBS. The organs were then homogenized by grinding. In the initial enrichment phase, we utilized Buffered Peptone Water (BPW, Haibo Biotechnology Co., Ltd., Qingdao, China), employing a 1:9 dilution ratio (sample in PBS: BPW). Subsequently, the composite was incubated at 37°C for 16–18 hr in a rotary incubator set to 180 rpm. For selective enrichment, Tetrathionate Broth Base (TTB, Land Bridge Biotechnology Co., Ltd., Beijing, China), fortified with iodine solution and brilliant green solution (both from Land Bridge Biotechnology Co., Ltd., Beijing, China), was employed at a ratio of 1:10 (sample in BPW: TTB). The mixture was subjected to incubation at 42°C for a duration ranging between 22 and 26 hr within a rotary incubator set at 180 rpm. Isolated *Salmonella* colonies from positive samples were obtained by subculturing selectively enriched samples on Xylose Lysine Deoxycholate (XLD, Land Bridge Technology Co., Ltd., Beijing, China) agar, followed by an 18–22 hr incubation at 37°C. Typical and pure colonies were selected after subculturing on XLD agar and transferred into Luria-Bertani broth. Finally, the transferred bacterial culture underwent an additional incubation for 18–22 hr at 37°C in a rotary incubator operating at 180 rpm.

## DNA extraction and genomic assembly

The DNA of 45 isolates was extracted using the Vazyme Fastpure Bacteria DNA Isolation Mini Kit (Vazyme Biotech Co., Ltd.), which was then quantified using the NanoDrop 1000 system (Thermo Fisher Scientific, USA). Subsequently, DNA libraries were constructed and subjected to sequencing using the Illumina NovaSeq 6000 platform (Beijing Novogene Co., Ltd.). Assembly of genome sequences was performed by SPAdes (*Bankevich et al., 2012*) v3.12.0 with default parameters.

## Global dataset assembly

To provide global context, we expanded the dataset by including 540 additional WGS data, consisting of bvSP (n=483), bvSG (n=50), bvSD (n=2), and *S. enterica* serovar Enteritidis (n=5). The inclusion of five strains of *S. enterica* serovar Enteritidis enhanced evolutionary analyses. Notably, in this collection of 540 genomic data from public databases, 325 sequences were previously preserved in our laboratory.

All WGS data passed strict quality control according to criteria set by the European Reference Laboratory (*Ellington et al., 2017*). Genomic data exceeding 500 contigs and having an N50 of less than 30,000 were excluded. Lastly, the bacterial species associated with each genomic dataset was confirmed by utilizing KmerFinder (*Hasman et al., 2014*) v3.2. Finally, the assembled database includes a total of 585 high-quality sequences.

## Genotyping and phylogenetic analyses

Snippy v.4.4.5 was used to identify cgSNPs, with the complete genome of R51 used as a reference. The optimal model (TVM+F) was determined and employed to construct the maximum likelihood phylogenetic tree using IQ-TREE v.1.6 (*Nguyen et al., 2015*). The final phylogenetic tree was annotated and visualized using iTOL v.6.0 (*Letunic and Bork, 2021*).

The global assembly of *S*. Gallinarum underwent a population structure analysis using the RhierBAPS v1.1.4 (*Cheng et al., 2013*), based on the hierarchical Bayesian clustering algorithm and employing cgSNPs as input. The calculation of population structure utilized R package v.4.3.1, supplemented by the R packages 'phytools' v.2.0.3, 'ape' v.5.7.1, and 'rhierbaps' v.1.1.4. Default parameters were employed for all population structure analysis.

## Temporal and phylogeographical analysis

To examine the emergence and geographical transfers, we first utilized Gubbins v.2.3 (*Croucher et al., 2015*) to eliminate the recombination regions for each lineages (*Figure 2—figure supplement 6*). Then, TreeTime (*Sagulenko et al., 2018*) was used to assess the temporal structure. This was accomplished through regression analysis of the root-to-tip branch distances within the maximum likelihood tree, considering the sampling date as a variable. Then, the strains within the dataset underwent phylogeographical reconstruction via Bayesian Evolutionary Analysis of Sampled Trees (BEAST) (*Bouckaert et al., 2019*) v.2.5. To determine the optimal model for running BEAST, we tested a total of six combinations in the initial phase of our study. These combinations included the strict clock, relaxed lognormal clock, and three population models (Bayesian SkyGrid, Bayesian Skyline, and Constant Size). Before conducting the full BEAST analysis, we evaluated each combination using a Markov Chain Monte Carlo analysis with a total chain length of 100 million and sampling every 10,000 iterations. We then summarized the results using NSLogAnalyser and determined the optimal model based on the marginal likelihood value for each combination. The results indicated that the model incorporating the Bayesian Skyline and the relaxed lognormal clock yielded the highest marginal likelihood value in our sample. Consequently, we proceeded to perform a time-calibrated Bayesian phylogenetic inference analysis for each lineage. The following settings were configured: the 'GTR' substitution model, '4 Gamma categories', the 'Relaxed Clock Log Normal' model, the 'Coalescent Bayesian Skyline' tree prior. Convergence was assessed using Tracer, with all parameter effective sampling sizes exceeding 200. Maximum clade credibility trees were generated using TreeAnnotator v.2.6.7. Finally, key divergence time points (with 95% credible intervals) were estimated and the phylogenetic tree was visualized using Figtree v.1.4.3.

## SNP distance-based *S*. Gallinarum geographical tracing

To understand potential transmission events, we calculated the cgSNP distances between distinct bacterial strains employing the SNP-dists. We estimated the overall evolutionary rate of the *S*.

Gallinarum using BEAST. We applied the methodology described previously (*Pightling et al., 2022*). The numbers of SNPs per year were determined by multiplying the evolutionary rates estimated with BEAST by the number of core SNP sites identified in the alignments. We hypothesize that a slower evolutionary rate in bacteria typically requires a lower SNP threshold when tracing transmission events using SNP distance analysis. Previous research found an average evolutionary rate of 1.97 SNPs per year (95% HPD, 0.48–4.61) across 22 different *Salmonella* serotypes. Our updated BEAST estimation for the evolutionary rate of *S.* Gallinarum suggests it is approximately 0.74 SNPs per year (95% HPD, 0.42–1.06). Based on these findings, as well as our previous experience with similar studies (*Feng et al., 2023*), we set a threshold of two SNPs (approximately representing less 2 years of evolution). Additionally, the geographical location was considered to enhance the precision of speculation regarding potential transmission sites and transmission direction.

## MGE detection

Four types of MGEs were detected: plasmids, transposons, integrons, and prophages. Plasmids were detected using Abricate v.1.0.1 with the PlasmidFinder (*Carattoli et al., 2014*) database. Only plasmids with a similarity of more than 90% and a coverage over 95% were identified. BacAnt (*Hua et al., 2021*) was used to detect integrons and transposons in the genomes. Only integrons or transposons with a similarity of more than 60% and a coverage greater than 60% will be identified. The prophages were detected using the Phaster pipeline. The genomic data were split into two temporary databases based on the number of contigs, one dataset containing single contig files and the other containing multiple contig files. The two databases were imported into the Phaster pipeline separately with default parameters.

## Determination of the MGE carried ARG and horizontal ARG transfer frequency

We have devised a pipeline to discern the coexistence of MGEs and ARGs. For plasmids and prophages, only ARGs situated within MGE regions are deemed MGE-carried. Regarding transposons and integrons, the search region was expanded to 5 kilobases upstream and downstream of ARGs, accounting for potential splicing errors. In this study, HGT events involving ARGs were defined as instances where ARGs exhibited perfect identity (100% coverage and 100% identity) and were located on MGEs across different strains. In a previous study (*Wang et al., 2024b*), horizontal transfer of ARGs was identified based on gene regions with 100% coverage and over 99% identity. We believe that these stricter criteria in our pipeline more accurately reflect the transfer of ARGs within the *S.* Gallinarum population. HGT frequency was utilized to evaluate the extent of horizontal transfer of ARGs as follows:

$$HGT\ frequency = \frac{\text{Horizontal ARG transfer count}}{\text{Total dissemination count}}$$

The pipeline developed in this study accepts ResFinder or Resistance Gene Identifier (RGI) results as input. Specific code and examples are uploaded to: https://github.com/tjiaa/Cal_HGT_Frequency, copy archived to *Jia, 2025*.

## Genomic data analysis

The serovars of all WGS data were confirmed through SISTR (*Yoshida et al., 2016*) v.1.1.0 and SeqSero2 (*Zhang et al., 2019*). Multilocus Sequence Typing was carried out using MLST v.2.22 with the senterica_achtman_2 scheme. ARGs were detected by ResFinder (*Zankari et al., 2012*). Both similarity and coverage were set to a minimum value of 90. The invasiveness index was calculated using methods previously reported (*Van Puyvelde et al., 2019*). Specifically, *Salmonella*'s ability to cause intestinal or extraintestinal infections in hosts is related to the degree of genome degradation. We evaluated the potential for extraintestinal infection by 45 newly isolated *S.* Gallinarum strains (L2b and L3b) using a model that quantitatively assesses genome degradation. We analyzed each sample using the 196 top predictor genes for measuring the invasiveness of *S.* Gallinarum, employing a machine learning approach that utilizes a random forest classifier and delta-bitscore functional variant-calling. This method evaluated the invasiveness of *S.* Gallinarum toward the host, and the distribution of invasiveness index values for each region was statistically tested using unpaired *t*-test. The code used for

calculating the invasiveness index is available at https://github.com/Gardner-BinfLab/invasive_salmo-nella (**Wheeler, 2023**).

## ARGs vertical evolution control

In this study, HGTphyloDetect pipeline (**Yuan et al., 2023**) was used to control for vertical evolution in the ARG sequences mentioned. We extracted base sequences for the eight ARGs as shown in *Figure 6b* with an HGT frequency greater than zero ($bla_{TEM-1B}$, *sul1*, *dfrA17*, *aadA5*, *sul2*, *aph(3")-Ib*, *tet(A)*, *aph(6)-Id*). For $bla_{TEM-1B}$, *sul1*, *dfrA17*, *aadA5*, and *sul2*, the HGT frequency reached 100% across different isolates, indicating that these ARG sequences have a unique ancestral sequence type. In contrast, due to the ResFinder settings requiring both similarity and coverage to meet a minimum value of 90%, the base sequences for *aph(3")-Ib*, *tet(A)*, and *aph(6)-Id* are not unique. Consequently, we applied the HGTphyloDetect pipeline individually to each sequence type of ARGs to verify their association with HGT events. Specifically, among 436 bvSP isolates collected in China, we identified two sequence types of *aph(3")-Ib*, four sequence types of *tet(A)*, and three sequence types of *aph(6)-Id*.

Subsequently, to identify potential ARGs horizontally acquired from evolutionarily distant organisms, we queried the translated amino acid sequences of each ARG against the National Center for Biotechnology Information (NCBI) non-redundant protein database. We then evaluated whether these sequences were products of HGT by calculating AI scores and out_perc values. The calculation of AI score is as follows:

$$AI\,score = \ln\left(bbhG + 1 \cdot 10^{-200}\right) - \ln\left(bbhO + 1 \cdot 10^{-200}\right)$$

In this study, bbhG and bbhO represent the E-values of the best blast hit in in-group and out-group lineages, respectively. The out-group lineage is defined as all species outside of the kingdom, while the in-group lineage encompasses species within the kingdom but outside of the subphylum. Regarding the calculation method for out_perc:

$$out\_pect = n_{outside\,kingdom}/n_{total\,hits}$$

Finally, according to the definition provided by the HGTphyloDetect pipeline, ARGs with AI score≥45 and out_perc≥90% are presumed to be potential candidates for HGT from evolutionarily distant species.

## Code availability

The open-source software used in this study includes:

BacAnt (https://github.com/xthua/bacant; **Hua, 2023**)
Abricate (https://github.com/tseemann/abricate; **Seemann and Grüning, 2020**)
Snippy (https://github.com/tseemann/snippy; **Seemann, 2020**)
SISTR (https://github.com/phac-nml/sistr_cmd; **Bessonov and Kruczkiewicz, 2020**)
MLST (https://cge.food.dtu.dk/services/MLST/)
SeqSero2 (https://github.com/denglab/SeqSero2; **Zhang, 2024**)
IQtree (https://github.com/iqtree/iqtree2; **Minh et al., 2025**)
BEAST (https://www.beast2.org)
SNP-dists (https://github.com/tseemann/snp-dists; **Seemann et al., 2021**)
Fast BAPS (https://github.com/gtonkinhill/fastbaps; **Tonkin-Hill, 2022**)
R (https://www.r-project.org; **R Development Core Team, 2023**)
Phaster (http://phaster.ca)
KmerFinder (https://bitbucket.org/genomicepidemiology/kmerfinder)
TreeTime (https://github.com/neherlab/treetime; **Neher et al., 2024**)

## Acknowledgements

This work was supported by the National Program on the Key Research Project of China (2022YFC2604201), the Zhejiang Provincial Natural Science Foundation of China (LZ24C180002; LR19C180001), the Hainan Provincial Joint Project of Sanya Yazhou Bay Science and Technology City

(2021JJLH0083), the Zhejiang Provincial Key R&D Program of China (2023C03045, 2022C02024), and the Open Project Program of the Jiangsu Key Laboratory of Zoonosis (R1902). We also thank Dr. Peide Li for providing the valuable resource for this investigation.

## Additional information

### Funding

| Funder | Grant reference number | Author |
|---|---|---|
| National Program on the Key Research Project of China | 2022YFC2604201 | Min Yue |
| Zhejiang Provincial Natural Science Foundation | LZ24C180002 | Min Yue |
| Hainan Provincial Joint Project of Sanya Yazhou Bay Science and Technology City | 2021JJLH0083 | Min Yue |
| Zhejiang Provincial Key R&D Program of China | 2023C03045 | Min Yue |
| Open Project Program of the Jiangsu Key Laboratory of Zoonosis | R1902 | Min Yue |
| Zhejiang Provincial Natural Science Foundation | LR19C180001 | Min Yue |
| Zhejiang Provincial Key R&D Program of China | 2022C02024 | Min Yue |

The funders had no role in study design, data collection and interpretation, or the decision to submit the work for publication.

### Author contributions

Chenghao Jia, Conceptualization, Data curation, Software, Formal analysis, Validation, Investigation, Visualization, Methodology, Writing - original draft, Writing – review and editing; Chenghu Huang, Haiyang Zhou, Data curation, Formal analysis; Xiao Zhou, Zining Wang, Abubakar Siddique, Xiamei Kang, Qianzhe Cao, Yingying Huang, Data curation; Fang He, Yan Li, Writing – review and editing; Min Yue, Conceptualization, Resources, Data curation, Supervision, Validation, Writing – review and editing

### Author ORCIDs

Chenghao Jia ⓘ https://orcid.org/0000-0003-4406-9213
Xiao Zhou ⓘ https://orcid.org/0000-0001-6510-4095
Zining Wang ⓘ https://orcid.org/0000-0001-7107-1613
Yan Li ⓘ https://orcid.org/0000-0003-4813-5783
Min Yue ⓘ https://orcid.org/0000-0002-6787-0794

Reviewer #1 (Public review): https://doi.org/10.7554/eLife.101241.4.sa1
Reviewer #2 (Public review): https://doi.org/10.7554/eLife.101241.4.sa2
Author response https://doi.org/10.7554/eLife.101241.4.sa3

## Additional files

### Supplementary files

Supplementary file 1. Information on 45 newly isolated *S.* Gallinarum biovar Pullorum (bvSP) originated from Yueqing and Taishun used in this study.

Supplementary file 2. Information on 540 *Salmonella* isolates was obtained from public sources to assemble the global database, with 325 sequences previously preserved in our laboratory.

Supplementary file 3. The regional classification of 436 *S*. Gallinarum biovar Pullorum (bvSP) strains isolated from China was conducted.

Supplementary file 4. Information on calculation of invasiveness index for 45 *S*. Gallinarum biovar Pullorum (bvSP) isolates newly originated from Yueqing and Taishun.

Supplementary file 5. SNP distance-based tracing analysis for the 95 strains from Zhejiang Province and those from China with available provincial information (n=435). Only strains with an SNP distance of two or fewer are considered likely to be involved in potential transmission events.

Supplementary file 6. Information on antimicrobial resistance genes carried by 528 *S*. Gallinarum biovar Pullorum (bvSP) isolates.

Supplementary file 7. Information on plasmids, transposons, integrons, and prophages carried by 528 *S*. Gallinarum biovar Pullorum (bvSP) isolates.

Supplementary file 8. A co-localization analysis was conducted to assess each antimicrobial resistance gene's (ARG)'s association with mobile genetic elements (MGEs). Among 621 ARGs identified in 436 *S*. Gallinarum biovar Pullorum (bvSP) isolates collected across China, 415 ARGs were found to be located on MGEs.

Supplementary file 9. Detection of horizontal gene transfer (HGT) of antimicrobial resistance genes (ARGs) carried by mobile genetic elements in *S*. Gallinarum biovar Pullorum (bvSP) genomes from China. Using the HGTphyloDetect pipeline, we calculated the Alien Index (AI) score and out_perc values for each ARG sequences. ARGs with AI score≥45 and out_perc≥90% were identified as potential candidates for horizontal ARGs transfer. Additionally, based on BLAST hit scores, we determined the most likely donor organisms for these ARGs.

Supplementary file 10. The horizontal gene transfer (HGT) frequency value for specific antimicrobial resistance genes was identified from *S*. Gallinarum biovar Pullorum (bvSP) isolated from different regions of China.

MDAR checklist

## Data availability

For the newly isolated 45 strains of *Salmonella* Gallinarum, genome data have been deposited in NCBI Sequence Read Archive (SRA) database. The "SRA Accession" for each strain are listed in ***Supplementary file 1***. Additionally, the genome data for the 540 publicly available genomes have been uploaded to figshare.

The following datasets were generated:

| Author(s) | Year | Dataset title | Dataset URL | Database and Identifier |
| --- | --- | --- | --- | --- |
| Chenghao J, Chenghu H, Haiyang Z, Xiao Z, Zining W, Abubakar S, Xiamei K, Qianzhe C, Yingying H, Fang H, Yan L, Min Y | 2024 | WGS data of 42 strains of *Salmonella* Gallinarum isolated from Taishun and Yueqing, Zhejiang Province, China | https://www.ncbi.nlm.nih.gov/bioproject/PRJNA1143713 | NCBI BioProject, PRJNA1143713 |
| Chenghao J, Chenghu H, Haiyang Z, Xiao Z, Zining W, Abubakar S, Xiamei K, Qianzhe C, Yingying H, Fang H, Yan L, Min Y | 2024 | 3 newly isolated S. Gallinarum from Taishun and Yueqing Raw sequence reads | https://www.ncbi.nlm.nih.gov/bioproject/PRJNA1176376/ | NCBI BioProject, PRJNA1176376 |
| Jia C | 2024 | Genomic data | https://doi.org/10.6084/m9.figshare.26028592 | figshare, 10.6084/m9.figshare.26028592 |

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
