## [Editor Report · eLife Assessment]

This **important** study analyzes a large dataset of *Salmonella* gallinarum whole-genome sequences and provides findings regarding the population structure of this avian-specific pathogen. The **convincing** results indicate regional adaptation of the mobilome-driven resistome and a role in the evolutionary trajectory of this pathogen that will interest microbiologists and researchers working on genomics, evolution, and antimicrobial resistance.

---

## [Referee Report · Reviewer #1 (Public review)]

Summary:

The investigators in this study analyzed the dataset assembly from 540 *Salmonella* isolates, and those from 45 recent isolates from Zhejiang University of China. The analysis and comparison of the resistome and mobilome of these isolates identified a significantly higher rate of cross-region dissemination compared to localized propagation. This study highlights the key role of the resistome in driving the transition and evolutionary history of S. Gallinarum.

Strengths:

The isolates included in this study were from 16 countries in the past century (1920 to 2023). While the study uses S. Gallinarun as the prototype, the conclusion from this work will likely apply to other *Salmonella* serotypes and other pathogens.

---

## [Referee Report · Reviewer #2 (Public review)]

Summary:

The authors sequence 45 new samples of S. Gallinarum, a commensal *Salmonella* found in chickens, which can sometimes cause disease. They combine these sequences with around 500 from public databases, determine the population structure of the pathogen, and coarse relationships of lineages with geography. The authors further investigate known anti-microbial genes found in these genomes, how they associate with each other, whether they have been horizontally transferred, and date the emergence of clades.

Strengths:

- It doesn't seem that much is known about this serovar, so publicly available new sequences from a high burden region are a valuable addition to the literature.

- Combining these sequences with publicly available sequences is a good way to better contextualise any findings.

- The genomic analyses have been greatly improved since the first version of the manuscript, and appropriately analyse the population and date emergence of clades.

- The SNP thresholds are contextualised in terms of evolutionary time.

- The importance and context of the findings are fairly well described.

---

## [Author Response]

The following is the authors’ response to the previous reviews.

**Public Reviews:**

**Reviewer #1 (Public review):**
Summary:The investigators in this study analyzed the dataset assembly from 540 *Salmonella* isolates, and those from 45 recent isolates from Zhejiang University of China. The analysis and comparison of the resistome and mobilome of these isolates identified a significantly higher rate of cross-region dissemination compared to localized propagation. This study highlights the key role of the resistome in driving the transition and evolutionary history of S. Gallinarum.Strengths:The isolates included in this study were from 16 countries in the past century (1920 to 2023). While the study uses S. Gallinarun as the prototype, the conclusion from this work will likely apply to other *Salmonella* serotypes and other pathogens.

Thank you very much for your positive feedback. We recognize, as you noted, that emphasizing *Salmonella enterica* Serovar Gallinarum in the title may lead readers to perceive our methods and conclusions as overly restrictive. In light of your evaluation of our work, we have revised the title to: “Avian-specific *Salmonella* transition to endemicity is accompanied by localized resistome and mobilome interaction” We believe this final version not only reflects the applicability of our conclusions, as you appreciated, but also addresses your previous suggestion to highlight the resistome and mobilome.

Revisions in the manuscript Lines: 1-3

Weaknesses:While the isolates came from 16 countries, most strains in this study were originally from China.

We believe that this issue was discussed in detail in our previous response. Although potential bias exists, we have minimized its impact by constructing the largest global S. Gallinarum genome dataset to date. In addition, we have further emphasized these limitations in the manuscript.

Comments on revisions:This reviewer is happy with the detailed responses from the authors regarding revising this manuscript. I do not have further comments.

We greatly appreciate your positive feedback and are pleased that our responses have addressed your concerns.

**Reviewer #2 (Public review):**
Summary:The authors sequence 45 new samples of S. Gallinarum, a commensal *Salmonella* found in chickens, which can sometimes cause disease. They combine these sequences with around 500 from public databases, determine the population structure of the pathogen, and coarse relationships of lineages with geography. The authors further investigate known anti-microbial genes found in these genomes, how they associate with each other, whether they have been horizontally transferred, and date the emergence of clades.Strengths:- It doesn't seem that much is known about this serovar, so publicly available new sequences from a high burden region are a valuable addition to the literature.- Combining these sequences with publicly available sequences is a good way to better contextualise any findings.- The genomic analyses have been greatly improved since the first version of the manuscript, and appropriately analyse the population and date emergence of clades.- The SNP thresholds are contextualised in terms of evolutionary time.- The importance and context of the findings are fairly well described.

Thank you so much for your thorough review and constructive comments on the manuscript.

Weaknesses:- There are still a few issues with the genomic analyses, although they no longer undermine the main conclusions:

We are grateful for the valuable time and effort you have dedicated to improving our manuscript. In this revision, we have provided a point-by-point response to each of your concerns. Moreover, with the addition of new supplementary materials and modifications to the figures, we have re-examined and adjusted the numbering of figures and supplementary materials in the text to ensure they appear correctly in the manuscript.

(1) Although the SNP distance is now considered in terms of time, the 5 SNP distance presented still represents ~7yrs evolution, so it is unlikely to be a transmission event, as described. It would be better to use a much lower threshold or describe the interpretation of these clusters more clearly. Bringing in epidemiological evidence or external references on the likely time interval between transmissions would be helpful.

We sincerely thank you for highlighting this issue. We appreciate your concern regarding the use of a 5-SNP threshold to define a transmission event, especially given the approximate 7-year evolutionary timeframe. Considering our updated estimate for the evolutionary rate of *S*. Gallinarum (approximately 0.74 SNPs per year, with a 95% HPD range of 0.42 to 1.06), we have revised the manuscript to use a 2-SNP threshold (approximately representing less than two years of evolution) to better control the temporal span of transmission events. In addition, we have updated the manuscript to reflect this new threshold and demonstrated that the use of a more stringent SNP threshold does not affect the overall conclusions of the study.

Specifically, we adopted the newly established 2-SNP threshold to update Figure 3a and corresponding Supplementary Figure 8. The heatmap on the far right of New Figure 3a illustrates the SNP distances among 45 newly isolated S. Gallinarum strains from two locations in Zhejiang Province (Taishun and Yueqing). New Supplementary Figure 8 simulates potential transmission events between the bvSP strains isolated from Zhejiang Province (n=95) and those from other regions of China with available provincial information (n=435). These analyses collectively demonstrate the localized transmission patterns of bvSP within China.

For New Figure 3a, we found that even with the 2-SNP threshold, the number of potential transmission events among the 45 newly isolated S. Gallinarum strains from the two Zhejiang locations (Taishun and Yueqing) remains unchanged. In fact, we observed that the results from SNP tracing using an SNP threshold of less than 5 are consistent (see Author response image 1).

**Author response image 1. sa3fig1:** Clustering results of 45 newly isolated *S*. Gallinarum strains using different SNP thresholds of 1, 2, 3, 4, and 5 SNPs. The five subplots represent the clustering results under each threshold. Each point corresponds to an individual strain, and lines connect strains with potential transmission relationships.

For New Supplementary Figure 8, we employed the 2-SNP threshold and found that the number of transmission events between the bvSP strains isolated from Zhejiang Province (n=95) and those from other Chinese provinces (n=435) decreased from 91 to 53. The names of the strains involved in these potential transmission events are listed in Supplementary Table 5.

Revisions in the manuscript

Lines: 352-357

Figures: Figure 3; Supplementary Figure 8

Table: Supplementary Table 5

(2) The HGT definition has not fundamentally been changed and therefore still has some issues, mainly that vertical evolution is still not systematically controlled for.

We sincerely thank you for highlighting this issue. We hope the following explanation will help clarify and improve our manuscript, as well as address your concerns.

In bacteria, mobile genetic elements (MGEs) such as plasmids, transposons, integrons, and prophages, as mentioned in our manuscript, are segments of DNA that encode enzymes and proteins responsible for mediating the movement of genetic material between bacterial genomes (commonly referred to as “jumping genes”). These MGEs contribute to the mechanisms of horizontal gene transfer (HGT) in *Salmonella*, including transduction (via prophages), conjugation (via plasmids), and transposition (via integrons and transposons) (*Nat Rev Microbiol.* 2005 Sep;3(9):722-32). These “jumping genes” can enable *Salmonella* to acquire additional antimicrobial resistance genes (ARGs), which may not only originate from other *Salmonella* strains but also from distantly related species.

To further address your concern regarding the systematic control of vertical evolution, we employed the HGTphyloDetect pipeline developed by Le Yuan et al. (*Brief Bioinform.* 2023 Mar 19;24(2):bbad035) to control for vertical evolution in the ARG sequences mentioned in our manuscript. We chose HGTphyloDetect because, as noted, "jumping genes" often occur among evolutionarily distant species, rendering the use of Gubbins potentially unsuitable for these distant HGT events.

Using the HGTphyloDetect pipeline, we extracted base sequences for the eight ARGs shown in Figure 6b with an HGT frequency greater than zero (*bla*^TEM-1B^, *sul1, dfrA17, aadA5, sul2, aph(3’’)-Ib, tet(A), aph(6)-Id*). For *bla*^TEM-1B^, *sul1*, *dfrA17*, *aadA5*, and *sul2*, the HGT frequency reached 100% across different isolates, indicating that these ARG sequences have a unique sequence type. In contrast, due to the ResFinder settings requiring both similarity and coverage to meet a minimum value of 90%, the base sequences for *aph(3’’)-Ib*, *tet(A)*, and *aph(6)-Id* are not unique. Consequently, we applied the HGTphyloDetect pipeline individually to each sequence type of ARGs to verify their association with HGT events. Specifically, among 436 bvSP isolates collected in China, we identified two sequence types of *aph(3’’)-Ib*, four sequence types of *tet(A)*, and three sequence types of *aph(6)-Id*.

Subsequently, to identify potential ARGs horizontally acquired from evolutionarily distant organisms, we queried the translated amino acid sequences of each ARG against the National Center for Biotechnology Information (NCBI) non-redundant protein database. We then evaluated whether these sequences were products of HGT by calculating Alien Index (AI) scores and out_perc values.

The calculation of AI score is as follows:=ln⁡(bbhG+1⋅10−200)−In⁡(bbhO+1⋅10−200)

In this study, bbhG and bbhO represent the E-values of the best blast hit in ingroup and outgroup lineages, respectively. The outgroup lineage is defined as all species outside of the kingdom, while the ingroup lineage encompasses species within the kingdom but outside of the subphylum. An AI score ≥ 45 is considered a strong indicator that the gene in question is likely derived from an HGT event.

Regarding the calculation method for out_perc:out_pect=noutside kingdom /ntotal hits 

Finally, according to the definition provided by the HGTphyloDetect pipeline, ARGs with AI score ≥ 45 and out_perc ≥ 90% are presumed to be potential candidates for HGT from evolutionarily distant species. We have compiled the calculation results for the aforementioned genes in New Supplementary Table 9. The results indicate that all ARGs presented in Figure 6b, which exhibited a HGT frequency greater than zero, were acquired horizontally by *S*. Gallinarum. Based on these findings, we have revised the manuscript accordingly.

Revisions in the manuscript

Lines: 302-307; 616-650; 955-957

Table: Supplementary Table 9

Using a 5kb window is not sufficient, as LD may extend across the entire genome.

We agree with your point that linkage disequilibrium (LD) could influence the transmission of genes within chromosomal regions. LD can lead to the non-random cooccurrence of alleles at different loci within a population. Considering that horizontal gene transfer (HGT) events involving more distantly related ARGs may be accompanied by vertical propagation on chromosomes, and to simultaneously assess the impact of LD, we conducted two evaluations.

It is important to note that the following assessments are based on the assumption that plasmid replicons detected by PlasmidsFinder are part of self-replicating, extrachromosomal DNA.

(1) In the revised pipeline used to calculate ARG HGT frequencies, we categorized a total of 621 ARGs carried by 436 bvSP isolates collected in China and found that 415 of these ARGs were located on MGEs. We further investigated the distribution of these 415 ARGs across different MGEs, taking into account the complex nesting relationships among them. We observed that 90% of the ARGs (372/415) were located on plasmid contigs. It is important to clarify that this finding does not contradict our statement in the manuscript regarding plasmids and transposons as the primary reservoirs for resistome geo-temporal dissemination. This is because transposons, integrons, and prophages carrying ARGs can also be found on plasmids. Additionally, only 25 bvSG isolates from China contained ARGs, which were likely acquired via transposons or integrons located on the chromosome.

(2) In our manuscript, we searched for ARGs within a 5kb upstream and downstream region (a total of 10kb) of transposons and integrons (The BLASTn parameters used in the Bacant pipeline to identify transposons and integrons were set to a coverage threshold of 60%, rather than 100%). However, in light of the potential impact of LD on vertical transmission, we expanded our search to include a 10kb upstream and downstream range (a total of 20kb) for these 25 isolates. The decision to expand the search range to 10kb upstream and downstream range is based on the following two considerations: (1) Based on literature, we determined the overall lengths of the integrons and transposons carried by the 25 isolates (Tn801, Tn6205, Tn1721, In498, In1440, In473, and In282), and found that the maximum length of these elements is ~13.5 kb. Using a 10kb upstream and downstream threshold effectively covers these integrons/transposons. (2) The limitation posed by genomic fragmentation due to next-generation sequencing, which restrict the search range. We present the results of this expanded search for colocalization of ARGs with transposons and integrons at: Figshare: https://doi.org/10.6084/m9.figshare.28129130.v1

We found that these results were consistent with those obtained using the previous search range.

Taken together, these results suggest that although linkage disequilibrium may influence genetic processes within chromosomal regions—particularly for the few chromosomeassociated antibiotic resistance genes linked to integrons and transposons—the overall impact in our study is likely minimal. This conclusion is supported by the observation that 90% of the ARGs in our dataset are located on plasmids, and even an expanded search range does not alter this outcome. Additionally, by incorporating Alien Index scores and calculating out_perc, we can further confirm the occurrence of horizontal gene transfer events.

However, it is undeniable that other studies using our current pipeline may be affected. As a temporary remedial measure, we have included a note in the "README" file as below (https://github.com/tjiaa/Cal_HGT_Frequency):

“Note: Considering that ARGs located on the chromosome and carried by mobile genetic elements—such as integrons and transposons—may introduce potential computational errors, we recommend evaluating the number of ARGs associated with these elements on the chromosome during your analysis. If a majority of ARGs in your dataset fall into this category, we suggest using additional methods to evaluate the potential impact of linkage disequilibrium. Additionally, by modifying the “MGE_start” and “MGE_end” parameters in the “eLife_MGE_ARG_Co_location.ipynb” script, you can assess the distance between different ARGs and integrons or transposons on the chromosome. This approach will further aid in evaluating the impact of linkage disequilibrium on the genetic process.”

We believe this approach will assist researchers in further assessing the potential impact of vertical evolution and help other users determine whether additional methods are necessary to account for such effects.

As the authors have now run gubbins correctly, they could use the results from this existing analysis to find recent HGT.

We sincerely thank you for your valuable suggestion. Utilizing additional methods to predict potential horizontal gene transfer (HGT) events could indeed enhance the robustness of the results. However, "jumping genes" often occur among evolutionarily distant species, rendering the use of Gubbins potentially unsuitable for these distant HGT events.

Furthermore, the primary focus of our study is to identify HGT of antimicrobial resistance genes (ARGs) in the *Salmonella* genome driven by mobile genetic elements. Therefore, we employed the HGTphyloDetect pipeline developed by Le Yuan et al. (*Brief Bioinform.* 2023 Mar 19;24(2):bbad035) to control for vertical evolution in the ARG sequences. The specific computational methods and conclusions have been detailed above.

To definite mobilisation, perhaps a standard pipeline such (e.g. https://github.com/EBIMetagenomics/mobilome-annotation-pipeline) would be more convincing.

Thank you for your valuable suggestion. We agree that defining mobilization using a standardized pipeline can add rigor and clarity to our analysis. The pipeline you referenced (https://github.com/EBI-Metagenomics/mobilome-annotation-pipeline) is an excellent resource and provides a robust approach to the identification and annotation of mobile genetic elements.

We have examined and run this pipeline, which uses “IntegronFinder” and “ICEfinder” to detect integrons, “geNomad” to identify plasmids, and “geNomad” and “VIRify” to detect prophages. Our initial checks revealed that the numbers of integrons, plasmids, and prophages identified using this pipeline were consistent with those detected in our study. However, due to the significantly different output formats, the results from this pipeline could not be integrated with the pipeline we used for calculating HGT frequency.

We will incorporate the standardized pipeline you suggested in future studies to further improve the reliability of our findings.

(3) The invasiveness index is better described, but the authors still did not provide convincing evidence that the small difference is actually biologically meaningful (there was no statistical difference between the two strains provided in response Figure 6). What do other *Salmonella* papers using this approach find, and can their links be brought in? If there is still no good evidence, a better description of this difference would help make the conclusions better supported.

We sincerely appreciate your thoughtful feedback. The initial introduction of the invasiveness index in our manuscript aimed to quantitatively assess the differences in invasiveness between two geographically distinct strains of *S.* Gallinarum (isolated from Taishun and Yueqing) by comparing the degradation of 196 top predicted genes associated with invasiveness in their genomes. We found a highly significant statistical difference (*P* < 0.0001) in the invasiveness index between them.

Several studies have also employed the invasiveness index to predict biological relevance in *Salmonella* strains, and we believe these examples provide further context for our approach:

(1) Caisey V. Pulford et al, Nat Microbiol, 2021, used the same method to calculate the invasiveness index for *Salmonella* Typhimurium and employed it to characterize the invasiveness of different lineage strains. They found that *Salmonella* in Lineage-3 exhibited the highest invasiveness index, suggesting an adaptation from an intestinal to a systemic lifestyle. The authors noted, "Although the invasiveness index cannot yet be experimentally validated, *Salmonella* isolates with different invasiveness indices produce distinct clinical symptoms in a human population (*BMC Med. 2020 Jul 17; 18(1):212*)". They emphasized the necessity of developing more robust methods to measure *Salmonella* invasiveness.

(2) Sandra Van Puyvelde et al, Nat Commun, 2019, reported that *Salmonella* Typhimurium sequence type 313 (ST313) lineage II.1 exhibited a higher invasiveness index compared to lineage II, suggesting that the two lineages might have distinct adaptations to an invasive lifestyle. Further experiments demonstrated significant differences between these lineages in terms of biofilm formation (A red dry and rough (RDAR) assay) and metabolic capacity for carbon compounds.

(3) Wim L. Cuypers et al, Nat Commun, 2023, calculated the invasiveness index for 284 global *Salmonella* Concord strains across different lineages and found that Lineage-4 potentially exhibited the highest invasiveness.

Given these evidences, we acknowledge that no significant difference in mortality was observed between the L2b and L3b *S*. Gallinarum strains in 16-day-old SPF chicken embryos. Existing literature suggests that strains with higher invasiveness indices may still exhibit differences in biofilm formation and metabolic capacities, reflecting their adaptation to different host environments. As such, we maintain that the invasiveness index remains a valuable metric for evaluating the genomic differences between *S*. Gallinarum strains from Taishun and Yueqing. We plan to further investigate these differences through phenotypic experiments in our next research.

In the revised manuscript, we have added the following discussion along with additional references:

Lines 358-365: “Moreover, the invasiveness index of bvSP from Taishun and Yueqing suggests that different lineages of S. Gallinarum recovered from distinct regions may exhibit biological differences. Previous studies have shown that strains with higher invasiveness indexes tend to be more virulent in hosts (30, 31), potentially causing neurological or arthritic symptoms in S. Gallinarum infections. Furthermore, strains with varying invasiveness indexes have been confirmed to differ in their biofilm formation abilities and metabolic capacities for carbon compounds (32).”

Revisions in the manuscript:

Lines: 358-365, 806-827.

In summary, the analysis is broadly well described and feels appropriate. Some of the conclusions are still not fully supported, although the main points and context of the paper now appear sound.

Thank you so much for your positive evaluation of our work. We hope that the revised manuscript meets your expectations and offers a more accurate interpretation of our findings.

**Recommendations for the authors:**

**Reviewer #2 (Recommendations for the authors):**
This is a great improvement over the first version and I thank the authors for a thorough response, as well as changing their conclusions in response to their improvements.Other small remaining issues:Figure 3: Heatmap of SNPs is hard to read in grayscale. It also just represents the between clade distances already shown by the tree. It would be more useful to present intraclade distances only to see the SNP resolution _within_ each lineage. Using a better colour scheme would also help.

Thank you for your insightful comments and suggestions regarding Figure 3. We agree that the grayscale heatmap may present challenges in terms of visual clarity. To address this, we have updated the heatmap with a more distinct color gradient, ensuring better contrast and easier interpretation (New Figure 3).

Regarding your second suggestion: "It would be more useful to present intraclade distances only to see the SNP resolution within each lineage," we believe it is already addressed in the current version of New Figure 3. Specifically, the heatmap on the right side of New Figure 3 illustrates the SNP distances between *S.* Gallinarum isolates from Taishun and Yueqing, with the goal of demonstrating that genomic variation within isolates from a single region is generally smaller compared to those from different regions. In this figure, 45 newly isolated *S.* Gallinarum strains are categorized into two lineages: L2b and L3b. The heatmap on the right side of Figure 3 displays the SNP distances between all pairwise combinations of these 45 strains, where the intraclade distances are represented by the red regions (highlighting the pairwise distances within each lineage, specifically L3b and L2b, which are indicated by two triangles). The between-clade distances are shown by the blue regions.

We also believe in further exploring the intraclade distances across the entire dataset of 580 *S.* Gallinarum strains, as it could provide additional insights. However, this analysis would extend beyond the scope of the current section.

Revisions in the manuscript Line: 998

Figure: Figure 3

Please remove Figure 6c, it does not add anything to the paper and raises questions about performing this regression.

Thank you for pointing out this issue. We have removed Figure 6c and the corresponding description in the "Results" section from the manuscript (New Figure 6).

Revisions in the manuscript Lines: 316, 319, 1035-1041.

Figure: Figure 6

Again, thank you all for your time and efforts in reviewing our work. We believe the improved manuscript meets the high standards of the journal.